# COVID-19 and Thymoquinone: Clinical Benefits, Cure, and Challenges

**Shimaa Abd El-Salam El-Sayed** [1,*,†] and **Mohamed Abdo Rizk** [2,†]

1   Department of Biochemistry and Chemistry of Nutrition, Faculty of Veterinary Medicine, Mansoura University, Mansoura 35516, Egypt
2   Department of Internal Medicine and Infectious Diseases, Faculty of Veterinary Medicine, Mansoura University, Mansoura 35516, Egypt
*   Correspondence: shimaa_a@mans.edu.eg
†   These authors contributed equally to this work.

**Abstract:** In today's world, the outbreak of the coronavirus disease 2019 (COVID-19) has spread throughout the world, causing severe acute respiratory syndrome (SARS) and several associated complications in various organs (heart, liver, kidney, and gastrointestinal tract), as well as significant multiple organ dysfunction, shock, and even death. In order to overcome the serious complications associated with this pandemic virus and to prevent SARS-CoV-2 entry into the host cell, it is necessary to repurpose currently available drugs with a broad medicinal application as soon as they become available. There are several therapeutics under investigation for improving the overall prognosis of COVID-19 patients, but none of them has demonstrated clinical efficacy to date, which is disappointing. It is in this pattern that *Nigella sativa* seeds manifest their extensive therapeutic effects, which have been reported to be particularly effective in the treatment of skin diseases, jaundice, and gastrointestinal problems. One important component of these seeds is thymoquinone (TQ), which has a wide range of beneficial properties, including antioxidant and anti-inflammatory properties, as well as antibacterial and parasitic properties, in addition to anticarcinogenic, antiallergic, and antiviral properties. This comprehensive review discussed the possibility of an emerging natural drug with a wide range of medical applications; the use of TQ to overcome the complications of COVID-19 infection; and the challenges that are impeding the commercialization of this promising phytochemical compound. TQ is recommended as a highly effective weapon in the fight against the novel coronavirus because of its dual antiviral action, in addition to its capacity to lessen the possibility of SARS-CoV-2 penetration into cells. However, future clinical trials are required to confirm the role of TQ in overcoming the complications of COVID-19 infection.

**Keywords:** COVID-19; thymoquinone; complications

## 1. Introduction

SARS-CoV, MERS-CoV, and the most recent 2019-nCoV, or SARS-CoV-2, are three large outbreaks of the coronavirus, a zoonotic virus known to cause respiratory disease, which have been recorded since 2002 [1]. The principal animal reservoir for coronaviruses is believed to be bats. However, in recent decades, the virus has been able to mutate and adapt to infect humans, resulting in a species barrier jump from animals to humans [2]. More than 200 countries have been affected by the new Covid sickness 2019 (COVID-19) pandemic, which was triggered by a severe respiratory disorder associated with Covid 2 (SARS-CoV-2) infection [3]. There are over 500 million individuals infected, with a loss of life more prominent than 6,248,873, and these numbers are growing [4], according to the World Health Organization. The severity of the symptoms experienced by COVID-19 patients vary, and they include dry cough, fever [5], sore throats, exhaustion, diarrhea, difficult breathing, and myalgia, as well as some biochemical abnormalities. Furthermore,

cardiovascular manifestations [6,7], such as acute cardiac injury, myocarditis, arrhythmia, and cardiovascular thromboembolism, have been found to be frequently associated with COVID-19 patients [5]. Moreover, neurological manifestations, such as dizziness, headache, loss of taste and smell, or reduced consciousness, have been observed in a high proportion of COVID-19 patients [8]. In severe cases, complications can occur within a few hours, including acute lung injury (ALI) and acute respiratory distress syndrome (ARDS), caused by the release of a large amount of pro-inflammatory mediators, such as interferon (IFN-), interleukin (IL-1b), tumor necrosis factor (TNF-), transforming growth factor (TGF), and chemokines, as a result of a cytokine storm initiated by the immune system. In severe cases of SARS-CoV-2 infection, this may result in organ failure and even death. Given the tragic reality that COVID-19 cannot be successfully treated, there is an urgent need for medications that are effective against SARS-CoV-2 infection. The development of prospective inhibitors from already-approved pharmaceuticals is crucial for the treatment of COVID-19 because it takes a while for new therapies to reach the market [9]. Plants are one of the most prevalent sources of the synthetic mixtures that are utilized as traditional medicines for human health. *Nigella sativa* (of the family *Ranunculaceae*) is commonly known as dark cumin, fennel bloom, or nutmeg flower [3]. Nigella sativa seeds are consumed for food in nations throughout the Middle East. To add flavor to bread or curries, it is typically lightly roasted, ground, or used whole. The seeds can also be consumed uncooked or combined with water or honey by certain people. They can also be included in yogurt, smoothies, and porridge [2,3]. For the first time, El–Dakhakhny [10] determined that the biological activities of Nigella sativa seeds—such as antioxidant; anti-inflammatory; antibacterial; antifungal; anti-viral; anti-parasitic and anti-protozoal; cytotoxic; anticancer; and neuro-, gastro-, cardio-, hetapto-, and nephro-protective activities—can be attributed to their basic oil composition, which is thymoquinone (TQ). TQ oil constitutes approximately 30–59% of the seed [10]. Treatment with TQ is effective in the treatment of a variety of illnesses, including neurodegenerative disorders, coronary vein infections, and respiratory and urinary system diseases [11]. TQ has also been shown to be effective in the treatment of inflammatory, cancerous, bacterial, antimutagenic, and antigenotoxic conditions [12,13]. Importantly, there have been reports of specific antiviral effects of TQ against viruses, such as the hepatitis C virus [13], the H9N2 avian influenza virus [12], and Epstein-Barr virus [14]. Indeed, TQ showed a wide spectrum of favorable biological activities, the most prominent being antioxidant, anti-inflammatory, and antibacterial activities [15]. These wide biological activities may explain why TQ works in so many different ways. However, the mechanisms of this broad spectrum of activities are still unknown.

Different studies demonstrated that COVID-19 infection is more severe in fragile patients, suggesting how immunotherapy or chemotherapy in cancer patients and immuno-supressive therapy in organ transplant patients could increase susceptibility to COVID-19 infection [16]. In fact, even before the onset of acute respiratory distress syndrome, individuals with severe and critical COVID-19 have lymphocytopenia and T-cell fatigue, which can lead to viral sepsis and a higher fatality rate [17]. When cancer patients, who are frequently immunocompromised, are given immune-checkpoint inhibitors, including avelumab and durvalumab, which blocks proteins called checkpoints that are made by some types of immune system cells, such as T cells, some cancer cells demonstrate restoration of their antitumoral immune response. Furthermore, T-cell depletion is seen in virally infected mice and humans, which is similarly seen after SARS-CoV-2 infection. Importantly, when they are treated with anti-cell surface receptor programmed cell death 1 (PD1) checkpoint protein and anti- programmed cell death-1 ligand 1 (PDL1) checkpoint protein antibodies, their T-cell competence is restored, and they are able to effectively combat viral infection [17]. Based on these findings, four clinical trials tested the efficacy of anti-PD1 antibody delivery in COVID-19-affected cancer and non-cancer patients [18]. In the same pattern, the morbidity and mortality rates were increased in COVID-19 patients suffering from fragile cardiovascular and respiratory diseases [19]. Interestingly, previous studies demonstrated the safety of TQ in the treatment of different diseases [20]. In light of

the safety and extensive therapeutic potential of TQ in cardiovascular disease, respiratory disease, and neurodegenerative disease, as well as its specific antiviral efficacy, we have decided to discuss the possibility of using TQ as an emerging natural drug to alleviate the complications associated with COVID-19 and to prevent SARS-CoV-2 entry into the host cell.

Five main steps are included in the life cycle of the coronavirus: attachment, penetration, biosynthesis, maturation, and release [3]. The main step is the attachment of the virus to the host cell, which is mediated through the Spike (S) protein, which consists of two subunits, S1 and S2, that target the human angiotensin-converting enzyme 2 (hACE2) receptor to bind to human cells in two stages [21]. The first stage depends on initiating the binding to the ACE2 receptor using the S1 subunit, and the second one is to mediate the membrane fusion with the S2 subunit [3]. The ACE2 receptor is highly expressed in the lung, heart, ileum, kidney, bladder, and epithelium, which explains the severe complications in these organs after coronavirus infection [4]. The progression of different complications for severe SARS-CoV-2 infection results from oxidative stress and cytokine storm-inducing multiple organ dysfunction syndromes (MODS) [22]. The pathophysiology of the coronavirus and its related complications are summarized in Figure 1. The following review will discuss different COVID-19-associated complications and the possible curative effect of TQ.

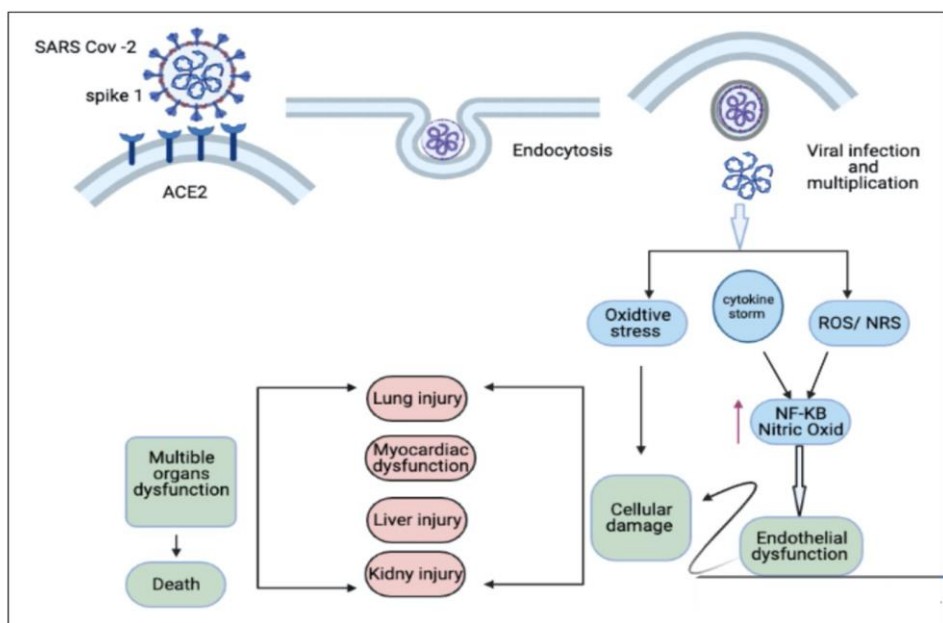

**Figure 1.** Entry of coronavirus and its associated complications. ACES (angiotensin-converting enzymes), NFKB (Nuclear factor-kappa), ROS (Reactive oxygen species), NRS (Nitrogen reactive species). Figure generated using BioRender.

## 2. Methods

Data of the current study were obtained from the most popular scientific databases, Web of Science (ISI), PubMed, Scopus, and Google Scholar, by searching keywords: 'COVID-19' and '*Nigella sativa*' or 'thymoquinone' and cardiopulmonary protective effect of TQ, hepatorenal effect of TQ or 'immunomodulatory effects' in the title or abstract. Relevant published articles in the English language up to February 2021 were included. All studies evaluating the effects of *N. sativa* or thymoquinone inflammatory lung diseases were included. Articles with insufficient information and in another language were excluded from the review.

### 3. Possible Curative Efficacy of TQ for the Inflammation and Multiple Organ Failure Associated with COVID-19

Coronavirus-induced cytokine release syndrome (CRS) is a serious condition that leads to multiple organ damage and Acute respiratory distress syndrome (ARDS) [23]. Once SARS-CoV-2 enters the host cell, it results in disruption of the intracellular environment through redistribution of the ion with activation of inflammation [24]. Two main substances—nucleotide-binding oligomerization domain (NOD), leucine-rich repeat (LRR9) and pyrin domain-containing protein 3 (NLRP3) and eicosanoids—are well known to play a critical role in inflammation, fever, and pain [25]. In addition, the disease is associated with increases in the secretion of proinflammatory cytokines: interleukins IL-1β, IL-18, IL-6, and tumor necrosis factor (TNF). Additionally, they have been found in critically ill individuals. This rise in proinflammatory cytokines is associated with the severity of the illness, and it is also a factor in the heightened cytokine storm and tissue inflammation that occur during respiratory illness [26]. Moreover, additional inflammatory mediators are involved in the pathogenesis of COVID-19, including Chemokines. CCL2 belongs to the group of CC chemokines and is also known as monocyte chemoattractant protein-1 (MCP-1) due to its participation in monocyte recruitment. It can bind to CC chemokine receptor type 2 (CCR2, CD192), triggering various downstream signaling pathways [27]. One of the most significant pathogenic outcomes of a severe SARS-CoV-2 infection is the infiltration of inflammatory monocytes and macrophages, as well as the dysregulated inflammation brought on by the function of these cells and the produced inflammatory mediators. However, the CCL2/CCR2 chemokine axis is essential for attracting and directing monocytes and macrophages to the lung tissue, according to [1]. Therefore, utilizing various medications to block this axis may lessen the severity of the condition and regulate excessive inflammation. Additionally, high plasma levels of several inflammatory mediators, including CCL2, granulocyte-macrophage colony-stimulating factor (GM-CSF), CXCL8 (interleukin-8), interferon gamma-induced protein 10 (IP-10), and osteopontin, were found in patients with SARS-CoV-2 infection, supporting the role of monocytes in the immunopathogenesis of COVID-19 [28]. Interestingly, according to studies, SARS-CoV-2 could infect mature cardiomyocytes, as well as those produced from human pluripotent stem cells, causing the release of CCL2 and the subsequent recruitment of monocytes. Monocyte infiltration and increased CCL2 expression were also found in the hearts of hamsters with SARS-CoV-2 infection [29].

Generally, COVID-19 therapy strategies target the viral replication cycle, which has been determined to be insufficient for increasing host survival and is also required to address the virus-induced cytokine release syndrome (CRS) [23]. Therefore, a drug that possesses the ability to inhibit both NLRP3 and eicosanoids is an urgent need. In this regard, TQ has been approved previously to have a promising anti-inflammatory role [22,30]. TQ acting as an NLRP3 inhibitor would consequently decrease secreted IL-1β, IL-18, and IL-6 and ameliorate pain and inflammation in COVID-19 patients. In addition to targeting NLRP3, TQ is also able to target the eicosanoid storm [31], which leads to inhibiting cytokine storm formation and, subsequently, helps prevent inflammation-mediated multiple organ damage in COVID-19 patients (Figures 2 and 3) (Table 1). One of the published clinical studies (NCT04401202) concluded that NSO (TQ) supplementation provides faster recovery of 62% of mild COVID-19 patients on day 14 of the treatment. The normal recovery time was also briefer than the control group. This study suggested that the reduction of COVID-19 symptoms (anosmia, chills, runny nose, and loss of appetite) might be due to the anti-inflammatory properties of NSO [32]. The potent anti-inflammatory effect of TQ, either in vitro or in vivo, together with its inhibitory effect on cytokine storm formation, highlights the possible curative efficacy of TQ on the inflammation and multiple organ failure associated with COVID-19.

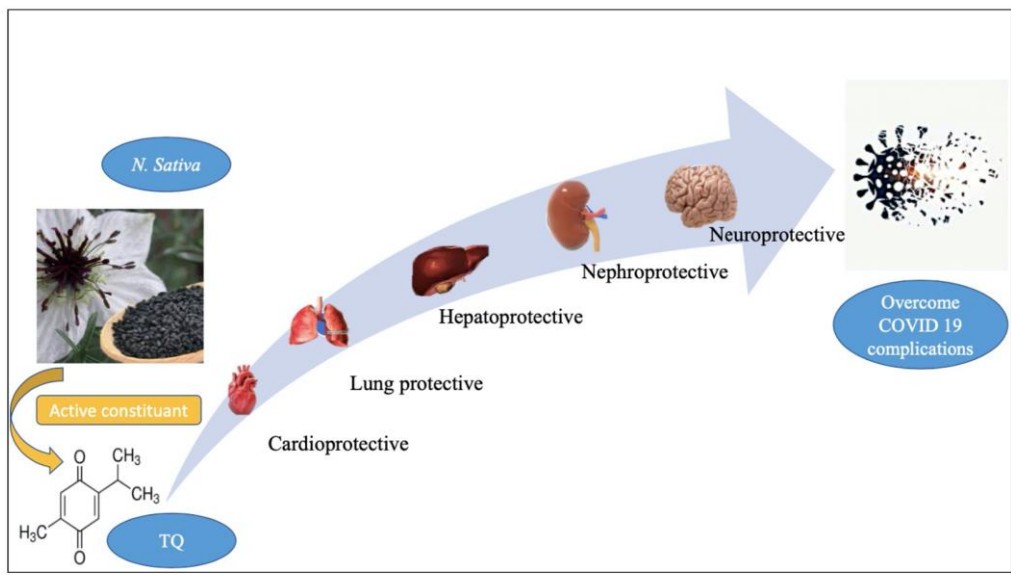

**Figure 2.** The general biological effect of thymoquinone (TQ). Figure generated using BioRender.

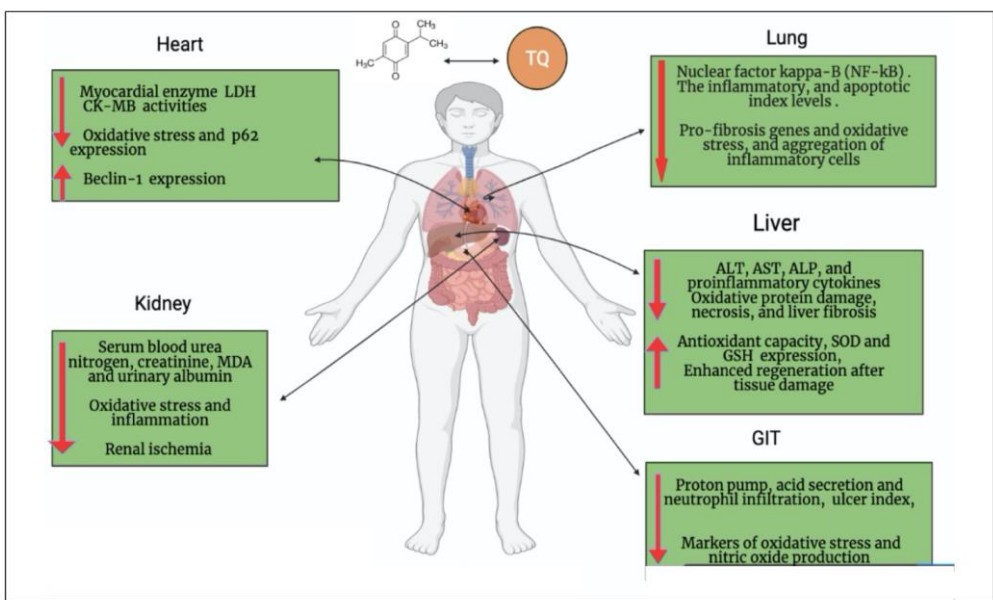

**Figure 3.** Possible curative efficacy of thymoquinone (TQ) on the complications associated with COVID-19 infection. Figure generated using BioRender.

## 4. Oxidative Stress Associated with COVID-19 and the Antioxidant Effect of TQ

It is well established that COVID-19 infection causes oxidative damage as a result of the oxidative stress the virus causes. Briefly, the basic mechanisms that regulate mitochondrial respiration and metabolism are disrupted as a result of interactions between some components of the reactive oxygen species (ROS) pathway and the proteins of the virus that infect cells [33]. According to reports, the severity of the condition was correlated with an increase in oxidative stress rates [34]. Therefore, it is advised to administer antioxidant supplements to lessen oxidative stress and the severity of the illness [34–36]. According to this trend, previous research showed TQ's antioxidant effects [6,37] (Figures 2 and 3) (Table 1). It was reported that TQ stimulates the expression of several detoxifying enzymes, including glutathione reductase, superoxide dismutase 1 (SOD1), catalase, and glutathione peroxidase 2 (GPX) [6,37,38]. A considerable rise in the level of antioxidant enzymes was observed in rats treated with TQ at a dose of 50 mg/kg body weight. The combination

of honey and NSO reduced COVID-19 symptoms, viral clearance, and mortality among COVID-19 patients, according to published results of a clinical trial (NCT04347382). In addition to existing COVID-19 therapies, this paper also promoted the usage of honey and NS [39]. According to this study, the combination of honey and NS provided its anti-COVID-19 activity due to their antioxidative/antiviral/immunostimulant chemical constituents (phenolic compounds, flavonoids, and zinc) that attack the multiple sites (lowering the expression of ACE-2 receptor, RdRp, Mpro protease, etc.) of SARS-CoV-2 [39]. Additionally, it has been suggested that N. sativa components may help cure COVID-19 by preventing the virus from entering the body, boosting the zinc immunological response against SARS-CoV-2, and preventing viral multiplication [40].

## 5. Cardiopulmonary Protective Effect of TQ

Cardiac injury is one of the fatal complications of COVID-19 [41]. Direct myocardial injury, systemic inflammation and a cytokine storm, downregulation of ACE2 receptors, abnormal myocardial oxygen demand-supply, plaque rupture with subsequent coronary thrombosis, side effects of several COVID-19 treatment options, electrolyte imbalances, and endothelial damage are some of the potential COVID-19 mechanisms that could result in CVD [42].

In a cohort investigation of subsequent autopsy cases conducted in Hamburg in April 2020, [43] detected the SARS-CoV-2 genome in the cardiac tissue in 24 out of 39 autopsies (61.6%). In addition to the virus's presence in the myocardial tissue and the progeny it produced, Linder et al. [43] also noted that the viral genome is not directly localized in the cardiomyocytes, but rather in the macrophages or interstitial cells that make up the cardiac tissue. Another study [44] proved that patients with cardiac injury had higher mortality when infected with COVID-19 than those without cardiac injury (42 of 82 [51.2%] vs. 15 of 334 [4.5%], respectively; $p < 0.001$). In a Cox regression model, patients with vs. those without cardiac injury were at a higher risk of death, both during the time from symptom onset (hazard ratio, 4.26 [95% CI, 1.92–9.49]) and from admission to end point (hazard ratio, 3.41 [95% CI, 1.62–7.16]). In addition, increasing the level of plasma troponin-T (c-Tnt) is one of the most important markers of cardiac damage [45]. Patients with severe COVID-19 instances had greater plasma levels of c-Tnt, required more mechanical breathing, were more prone to malignant arrhythmias, and required glucocorticoid medication [41]. Furthermore, cardiac damage in severe cases of COVID-19 is characterized by increasing and decreasing the expression of P62 and beclin1 in plasma, respectively [46]. Interestingly, TQ induces a cardioprotective effect through four main scenarios: (i) it significantly decreases *cardiac* troponin T (TnT) levels and markedly reduces cardiac tissue-inflammatory cell infiltration [47]; (ii) it decreases the expression of P62 and increases the expression of beclin1 [41]; (iii) it restores cardiomyocyte injury enzymes, leading to the repair of injured cardiomyocytes [48]; and (iv) it enhances the production of endogenous antioxidants and attenuates oxidative stress, which results in maintaining the structural integrity of myocardial muscle [6] (Table 1). Additionally, it is believed that TQ's potential for preventing CVD is a result of its ability to stimulate endothelial cells' production of NO and endothelium-derived hyperpolarizing factor (EDHF); decrease the endothelial production of vasoconstrictive factors, such as thromboxane A2; and also have an antioxidant effect on vascular SMCs. Therefore, TQ's effects on the endothelium and SMCs may improve vascular health in COVID-19 patients and may even lessen the disease's morbidity and mortality. Numerous studies have also demonstrated the effectiveness of TQ and NS seeds in preventing the production of thrombi [49]. It is a well-known fact that thrombus formation causes multiple organ collapse and fatality among COVID-19 patients. Therefore, NS may be used as a therapeutic formulation, including its nano-formulations, to treat COVID-19, and it may also be used as a supportive therapy with anti-COVID-19 medicines [49].

Breathlessness, pneumonia, and lung fibrosis are the main respiratory symptoms related to COVID-19 infection [50]. Acute SARS-CoV-2 infection results in denudation

of airway epithelial cells, with the accumulation of debris, which leads to obstructed airway functions and, subsequently, acute lung injury (ALI), as well as the more severe form, acute respiratory distress syndrome (ARDS) [50]. The ameliorating effect of TQ on respiratory disease and its promising effect on lung protection have been studied (Figures 2 and 3) (Table 1). In this regard, TQ decreases lung damage induced by long-lasting exposure to toluene in rats [51]. It also inhibits pulmonary fibrosis induced by bleomycin, lipopolysaccharide (LPS), and cyclophosphamide [52] through inhibition of activated NF-kβ in lung tissues, downregulated pro-fibrosis genes, decreased oxidative stress, and significantly reduced PGE2, TGF-β1, and INF-γ [53]. TQ (50 mg/kg b.w.) treatment significantly ($p < 0.05$) decreased the level of inflammatory cytokines, such as TNF-α, IL-1β, IL-6, and ICAM1, resulting from Benzopyrene toxication [54]. *N. sativa* oil exhibits airway anti-inflammatory and immune-regulatory effects, which may support its use for treatment of allergic asthma. Peripheral blood eosinophil count, IgG1 and IgG2a levels, cytokine profiles (IL-2, IL-12, IL-10, and IFN-γ levels), and inflammatory cells counts in lung tissue were significantly decreased by the plant oil in a mouse model of allergic asthma. The plant showed comparable immunomodulatory properties with dexamethasone, except that the plant had a greater effect on IFN-γ levels. Moreover, it was reported that *N. sativa* oil in the dose of (500 mg soft-gel capsules) one capsule orally, twice daily for 10 days, plus standard of care treatment has potential outcomes on patients with mild COVID-19 [55]. Another clinical study (IRCT20180712040449N2) was conducted in Iran, utilizing NS seed powder and a mixture of different herbs. This treatment significantly reduced the hospital dyspnea, accelerated recovery time, and lowered the COVID-19 symptoms. This study implicitly indicates that the chemical constituents of NS (TQ, hederagenin, THQ, nigelledine, and α-hederin) are anti-COVID-19 compounds.

## 6. Neuroprotective Effect of TQ can Overcome the Neurologic/Cognitive Manifestations Associated with COVID-19

Patients with COVID-19 infection suffer from different signs, such as headache, memory loss, mood changes, vision changes, hearing loss, impaired mobility, limb numbness, tremor, fatigue, and myalgia [56]. This is along with cases of encephalitis, necrotizing hemorrhagic encephalopathy, stroke, and epileptic seizures [57]. It is well known that neuroinflammation, induction of inflammation, and oxidative stress response are the main factors involved in the pathogenesis of almost all neurodegenerative diseases [58]. Subsequently, the administration of natural neuroprotective agents may reduce both neuroinflammation and oxidative stress, which may help in the recovery of COVID-19 patients [59]. In this regard, two previous studies [60] in rats exposed to lipopolysaccharides-induced neuroinflammation reported the effect of TQ in the inhibition of inflammatory mediators (TNF-a, IL-6, and IL-1beta) and their messenger RNA (mRNA) levels in BV2 microglia. Also, TQ may have the ability to ameliorate motor impairment and memory loss associated with COVID-19 infection, as it successfully inhibits rotenone-induced Parkinson's disease symptoms in animal models through stopped motor defects [61] and prevented neurotoxicity induced by amyloid protein (Ab1-42) in hippocampal and cortical neurons via ameliorating oxidative stress and improving the level of lipid peroxide changes in the hippocampal region, SOD, and acetylcholine esterase (AChE) activities [62]. The ameliorative effect of TQ on the inflammatory mediator, antioxidant enzymes, and neurotoxicity indicate the possible defensive effect of this natural compound against neurological complications associated with COVID-19 (Figures 2 and 3) (Table 1).

## 7. Hepatorenal Protective Effect of TQ against COVID-19

With the increasing number of COVID-19 infected patients, several studies reported that the liver is the most frequently affected organ after lung damage. Liver injury is a serious, fatal complication of COVID-19 [63]. The mechanism of hepatic injury in COVID-19 is not completely known. However, the injury may be caused directly by the invasion of the virus in the liver tissue, or it may be indirect (drug induced or the effect of inflammatory

mediators) [64], and the last one is more prominent. Moreover, patients suffering from parasitic infections, including malaria, Schistosoma, and Fasciola with increasing liver fibrosis and liver injury, are more susceptible to COVID-19 liver complications. A previous study has shown that schistosomiasis and helminth infection may increase the rate of unfavorable COVID-19 pandemic outcomes [65].

Helminth infections are typically connected with Th2-mediated immune responses [65]. Commonly, schistosomiasis infection leads to downregulation of the inflammation associated with Th2 immune response and subsequently lowers immunity to COVID-19, with increased susceptibility and higher incidence of COVID-19 in schistosomiasis-endemic areas of Africa [65]. Therefore, a treatment that ameliorates liver fibrosis of different origins and improves hepatotoxicity is required. In previous years, several studies reported the hepatoprotective effect of TQ, especially on liver toxicity and fibrosis, and their consequences [66,67]. Administration of TQ protects against hepatotoxicity associated with chemotherapy by reducing liver injury markers (SGOT: serum glutamic-oxaloacetic transaminase, SGPT: Serum glutamic pyruvic transaminase, GGT: gamma-glutamyl transferase) and tumor marker (alphafetoprotein) expression [68]. The hepatoprotective role of TQ may be attributed to its strong antioxidant property. Furthermore, TQ maintains the normal level of intracellular enzymes (reduced glutathione) and keeps the integrity of the membrane by reducing the leakage of AST and ALT [69]. TQ also reduced the damage to a liver cell and accumulation of extracellular matrix proteins, such as collagen, tenascins, laminins, and elastin. TQ was found to overcome liver fibrosis by reducing the mRNA levels of α-smooth muscle actin (α-SMA), collagen-I, and tissue inhibitor of metalloproteinase-1 (TIMP-1) [67]. Moreover, another study attributed the activity of TQ to improving liver function and the immunological system of infected mice, and partly to its antioxidant effects [69] (Figures 2 and 3) (Table 1). Subsequently, TQ can overcome hepatic injury associated with COVID-19.

Kidney injury is another COVID-19-related serious complication [70]. Certain chemotherapeutic regimens used for the treatment of COVID-19 result in nephrotoxicity [71]. TQ shows protective effects on the kidneys against mercuric chloride-induced renal damage [72]. TQ enhanced kidney function indicators, including blood urea nitrogen and creatinine, in addition to ameliorating antioxidant enzymes (GSH level and activities of GSHPx and CAT) in the renal cortex with inhibited lipid peroxidation [73]. Moreover, TQ shows reno-protective effects in sepsis-induced acute kidney injury (AKI). AKI is mediated by dysregulated activation of inflammasomes and proinflammatory cytokines that can be ameliorated by anti-inflammatory properties of TQ [74], where TQ decreases apoptosis of kidney cells and alleviates AKI. TQ supplementation improved the sloughing off of epithelial cells, contraction of glomeruli, and necrosis of renal tubules induced by cypermethrin in the kidneys of mice [75]. Finally, TQ reverses increased NFκB expression in the kidney of septic mice [74]. Collectively, by controlling pyroptosis, proinflammatory cytokines, and apoptosis-related expression, TQ lessens sepsis-induced AKI and ameliorates kidney damage following COVID-19 infection [74] (Figures 2 and 3) (Table 1).

## 8. Gastrointestinal Dysfunction Associated with COVID-19 and the Gastroprotective Effect of TQ

It has been reported that patients with COVID-19 experience diarrhea, as well as nausea/vomiting and abdominal pain [76]. In addition to acting as a gastroprotective agent, TQ also acts as a proton pump inhibitor and increases mucin secretion [77] (See Figures 2 and 3 for examples) (Table 1). As a possible complication of COVID-19, commensal bacteria in the gastrointestinal tract (GIT) may cause secondary bacterial infection. As a result of this pattern, the administration of TQ, which has a broad spectrum of antibacterial efficacy, especially against Gram positive cocci (*Staphylococcus aureus* ATCC 25923 and *Staphylococcus epidermidis* CIP 106510), is advised. It is possible that the antibacterial effect of TQ is due to: (a) increased ROS enervation, which is responsible for cell death caused by oxidative stress; (b) TQ inhibits biofilm formation and, as a result, inhibits its binding and

matrix formation, resulting in changes in the phenotype of the organisms due to changes in growth rate and gene transcription; or (c) TQ has specific selective cytotoxicity towards bacterial cells without causing membrane damage to normal cells [78].

**Table 1.** The beneficial effects of TQ against COVID-19 pathophysiological effects.

| COVID-19 Complications | Thymoquinone | References |
|---|---|---|
| Inflammation and cytokine release syndrome (CRS) | - NLRP3 inhibitor<br>- Decrease the secreted IL-1β, IL-18, and IL<br>- Ameliorate pain and inflammation<br>- Eicosanoid storm inhibitor | [22,30–32] |
| Oxidative damage | - Stimulates the expression of several detoxifying enzymes<br>- Increase glutathione reductase-Increase superoxide dismutase 1 (SOD1)<br>- Stimulate catalase and glutathione peroxidase 2 | [39,40] |
| Cardiac injury | - Decreases *Cardiac* troponin T (TnT) levels<br>- Reduce cardiac tissue-inflammatory cell infiltration<br>- Decrease the expression of P62 and increase the expression of beclin1<br>- Restore cardiomyocyte injury enzymes<br>- Enhance the production of endogenous antioxidants<br>- Attenuate the oxidative stress -Increase production of NO and endothelium-derived hyperpolarizing factor (EDHF),<br>- Decrease the endothelial production of vasoconstrictive factors like thromboxane | [41,47–49] |
| Pulmonary damage | - Inhibits pulmonary fibrosis<br>- Inhibition of activated NF-kβ in lung tissues<br>- Downregulated pro-fibrosis genes<br>- Decreased oxidative stress<br>- Reduced PGE2, TGF-β1, and INF-γ<br>- Decrease Peripheral blood eosinophil count, IgG1 and IgG2a levels<br>- Decrease cytokine profiles (IL-2, IL-12, IL-10, and IFN-γ levels) and inflammatory cells counts in lung tissue | [53–55] |
| Neurological disease | - Inhibits inflammatory mediators (TNF-a, IL-6, and IL-1beta) and their messenger RNA (mRNA) levels in BV2 microglia metabolites in the brain- Ameliorate motor impairment and memory loss<br>- Ameliorating oxidative stress and improving the level of lipid peroxide changes in the hippocampal region<br>- Decrease SOD and acetylcholine esterase (AChE) activities | [60–62] |

Table 1. *Cont.*

| COVID-19 Complications | Thymoquinone | References |
|---|---|---|
| Liver injury | - Reducing liver injury markers (SGOT, SGPT, and GGT)<br>- Maintains the normal level of intracellular enzymes (reduced glutathione)<br>- Keeps the integrity of the membrane by reducing the leakage of AST and ALT<br>- Retreated the damage to a liver cell and accumulation of the extracellular matrix proteins, such as collagen, tenascins, laminins, and elastin.<br>- Overcome liver fibrosis by reducing the mRNA levels of α-smooth muscle actin (α-SMA), collagen-I, and tissue inhibitor of metalloproteinase-1 (TIMP-1). | [67–69] |
| Kidney injury | - Enhanced the kidney function indicators, including blood urea nitrogen and creatinine<br>- Ameliorating the antioxidant enzymes (GSH level and activities of GSHPx and CAT) in the renal cortex with inhibited lipid peroxidation.<br>- Decreases apoptosis of kidney cells<br>- Improved the sloughing off the epithelial cell, contraction of glomeruli, and necrosis of renal tubules<br>- Reverses the increased NFκB expression in the kidney of septic mice | [72,74,75] |
| GIT injury | - Increased ROS enervation, which is responsible for cell death caused by oxidative stress,<br>- Inhibits biofilm formation and, as a result, inhibits its binding and matrix formation<br>- Changes in the phenotype of the organisms<br>- Changes in growth rate and gene transcription<br>- Specific selective cytotoxicity towards bacterial cells without causing membrane damage to a normal | [78] |

## 9. Thymoquinone Block SARS-CoV-2 Entry into Cells

Several natural compounds have been identified to inhibit COVID-19 infection. Quercetin, which is a constituent of apples, honey, raspberries, onions, and red grapes, shows antioxidant and anti-inflammatory, anti-cancerous, anti-viral, anti-bacteria, and immune modulatory effects, and it has the ability to inhibit 3CL protease activity and viral entry of SARS-CoV-2 inside the host cell. Caffeic acid inhibits the virus attachment to the host cell and binds 3CL protease, resulting in inhibition of the viral replication. Moreover, thymol, which is extracted from Thymus vulgaris, Ocimum, Origanum, inhibits the viral spike protein, prevents SARS-CoV-2 entry, and has a potent disinfectant effect. Ellagic acid, which is found in raspberries, strawberries, pomegranate, persimmon, grapes, and black currants, can inhibit COVID-19 through inhibition of the M$^{pro}$ and RdRp, and it prevents viral attachment and internalization to the host cell [79].

The promising efficiency of TQ in successfully treating different complications associated with COVID-19 infection and its long history of successful antiviral activity against the hepatitis C virus [12], H9N2 avian influenza virus [13], Epstein–Barr virus [14], and cytomegalovirus [14] attracts the attention of researchers to study the ability of TQ to inhibit the entry and survival of SARS COVID-19 in different organs cells. In this regard, PSH-DFK and Naber [80] proved not only the ability of TQ to inhibit the entry and survival of SARS-CoV-2 in a live cell, but also its merit over commonly used drugs in inhibiting the entry of SARS-CoV-2, including chloroquine (CQ) and hydroxychloroquine (HCQ). CQ and HCQ have a temporary inhibitory effect on the entry of SARS-CoV-2 through

changing the pH of the perfusion fluid. The issue can be reversed when either of the drugs is removed, while TQ increases endosomal pH, preventing SARS-CoV-2 entry into the cell, together with simultaneously attacking the virus due to the two single oxygens in the TQ molecule, thereby acting as both a shield and sword [81]. Moreover, the cationic amphiphilic nature of CQ and HCQ [82] results in their immobilization in the hydrophilic environment through the body, with the inability to reach the target organs. In contrast, the hydrophobic nature of TQ and its smaller size relative to CQ and HCQ protect it from early immobilization during its passage to the target organs and facilitates its cross through the plasma membrane of infected cells. Moreover, the hydrophobic nature of TQ contributed to its merit in destroying SARS-CoV-2 before entering cells, simply by binding to the lipophilic envelope of the virus, in agreement with the hydrophobic nature of the compound, and by oxidizing it [80]. In addition, consider the success of the treatment in quickly accessing a target through a viscous medium. Thus, the smaller the size of a molecule, the easier it can diffuse in a viscous medium. In this regard, it was reported that TQ precedes the other constituents of *N. sativa* seeds (nigellone molecule) due to its small size compared to its relatively large size. Subsequently, the time for TQ to reach a pathogen attached to or incorporated into a cell is expected to be shorter than for nigellidine and a-hederin, and, thus, it could be considered that TQ is a faster virus killer with superior residence time in the body, and it can achieve a deeper penetration into the tissue than nigellidine and a-hederin [80].

On the contrary, recent studies have reported the ability of TQ to hinder the binding between SARS-CoV-2 and ACE2 or cell surface heat shock protein (HSPA5) receptors and subsequently inhibit the entry of the virus into the cell [83] and reduce the risk of infection [84]. In addition, [85] has proven the noteworthy antiviral activity of TQ against a SARS-CoV-2 strain isolated from Egyptian patients by possibly preventing COVID-19 development by interacting with the receptor-binding domain on the spike and envelope proteins of SARS-CoV-2, which may obstruct virus entry into the host cell and inhibit its ion channel and pore-forming activity. Additionally, it has been found that TQ exhibits strong antagonistic activity against angiotensin-converting enzyme 2 receptors, allowing it to prevent virus uptake into the host cell. It may also have an inhibitory effect on SARS-CoV-2 proteases, which could reduce viral replication. Therefore, the movement of TQ from experimental application to clinical usage may be appropriate for testing against the COVID-19 pandemic.

## 10. Limitations of Clinical Application of TQ

In general, TQ toxicity is related to the type of metabolites that result from TQ metabolism inside the body and the lethal dose of 50% ($LD_{50}$) of TQ. El-Najjar et al. [86] reported that reductase enzymes can metabolize TQ into three different compounds, according to repeated reduction cycles. These compounds were the prooxidant semiquinone, antioxidant thymohydroquinone [86], and dihydro-thymoquinone. Due to its high redox activity, dihydro-thymoquinone can undergo a redox cycle with its semiquinone radical anions, which causes the creation of reactive oxygen species (ROS), such as superoxide, hydrogen peroxide, and finally the hydroxyl radical. The development of oxidized cellular macromolecules, such as lipids, proteins, and DNA, caused by ROS production can change the redox equilibrium inside cells and have a negative impact on the health of the cells [87]. The cytotoxicity and genotoxicity of TQ have been intensively studied in vitro [88], and the results showed that TQ with high concentrations ranging from 10 μM to 50 μM has cytotoxic effects due to the induction of high levels of necroses, glutathione depletion, and liver damage in a concentration-dependent manner. Another cytotoxic effect was determined according to the $LD_{50}$ of TQ administered via different routes. Previous reports indicate that $LD_{50}$ of TQ via the oral route is 2.4 g/kg in mice, according to Badary [89], and 870.9 mg/Kg, according to Al-Ali et al. [90]. This amount showed hypoactivity, trouble in the breath, a critical decrease in tissue (liver, kidneys, and heart), and diminished GSH content [89]. Moreover, $LD_{50}$ of TQ administered I/P in mice is 90.3 mg/kg [91] and

104.7 mg/kg, according to Al-Ali et al. [90], and its cytotoxic effects include an increase in malondialdehyde and catalase activities, even at a dose of 40 mg/kg [92]. In the rat, LD50 of TQ was 57.5 mg/kg i.p. and 794.3 mg/kg p/o, according to Al-Ali et al. [90]. This toxicological aspect of TQ hinders its clinical application and impedes its wide therapeutic usage. Therefore, these issues require urgent development of new strategies to overcome TQ toxicity and facilitate its use. In the next section, we will talk about the suggested methodologies to ameliorate TQ cytotoxicity.

## 11. New Insights into the Future of TQ Clinical Application for Treatment of COVID-19

Despite the conspicuous impact and wide biological activities of TQ on various sicknesses, its application has been restricted to exploratory models, which dismissed the clinical assessment of TQ viability, either in animal or human subjects. The translation of the experimental findings into reality is the key factor for the successful application of certain studied drug candidates, and this depends on clinical trials to formulate it into the traditional common forms, such as tablets and capsules [93]. In the last few years, several trials have been applied to develop the pharmaceutical formulation of TQ [94]. However, TQ possesses several toxic effects and physical drawbacks that hinder its pharmaceutical formulation [93], including low stability in aqueous solutions, as an alkaline solution can cause rapid degradation of TQ, as well as its hydrophobic nature, with low water solubility (549–669 μg/mL) and, consequently, low bioavailability [95]. Moreover, TQ is highly sensitive to light and temperature, in addition to non-targeted drug distribution [96]. In this regard, the development of new strategies that maintain the potential therapeutic effect and safety of TQ, with an increase in its clinical preparation and bioavailability, is an important demand [94].

For reducing toxicity, a technology of enveloping TQ nanoparticles into nanocarrier materials has been developed [97]. In this technology, the used nanocarrier materials are synthesized in a hydrophilic colloidal system, thus allowing the formation of water-born TQ nanoparticles by successfully rendering the hydrophobicity of TQ and increasing its bioavailability [97,98]. The previous investigation compared the toxicity of TQ administered orally, before and after encapsulation in lipid nanocarriers [99], and reported that encapsulated TQ has lower toxicity than the non-capsulated form, with no side effects after prolonged oral administration with a dose of 10 mg/kg mice/day [99] and 0.813 mg/kg human body weight/day [100]. The different carrier materials that can be used in the synthesis of TQ nanoparticles are as follows: (i) nanoparticles with synthetic polymer, such as polyethylene glycol [101] and polylactide coglycolide [102]; (ii) polymeric micelle, such as Pluronic F127 and Pluronic F68 [103]; (iii) nanoparticles with natural polymer, such as chitosan [104]; (iv) nano-lipid carriers, including nanoliposomes [105] and solid lipid nanoparticles [106]); (v) nano emulsions [96,107]; (vi) microemulsions [108]; and (vii) silica nanoparticles with impregnated TQ, rather than encapsulated [109]. Thymoquinone drawbacks and possible strategies to overcome them are summarized in Figure 4.

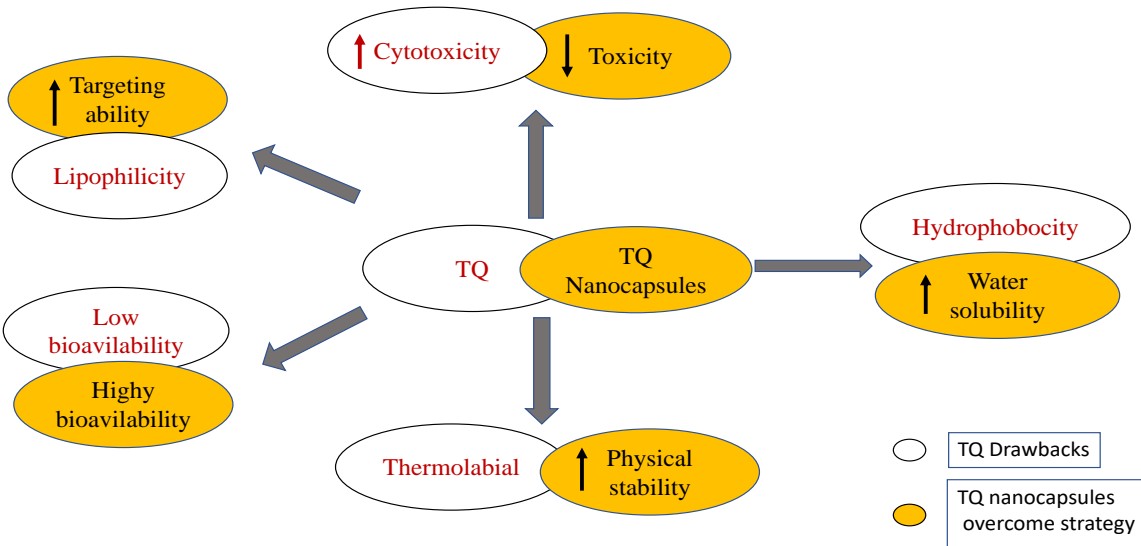

**Figure 4.** Thymoquinone drawbacks and the suggested strategies to overcome the drawbacks. Figure generated using BioRender.

One of the most important aspects of the pharmaceutical formulation of TQ into conventional dosage formats, such as tablets and capsules, is the successful production of TQ in large quantities to satisfy demand. Normally, TQ is obtained from the volatile oil fraction of N. sativa seeds, which represents around 67% of the volatile oil composition of high-quality seeds [110]. Two techniques have been developed to extract pure TQ from the seeds. The first relies on mixing n-hexane with the whole crude oil of the seed to separate TQ [110], followed by profound freezing at −20° [111]. The other technique is based on nanotechnology methods to separate TQ from N. sativa volatile oil [112].

Despite the continuous efforts to produce pure TQ from natural sources, the optimal obtained amount is not sufficient to satisfy the large demand for TQ on a worldwide level due to the unavailability of natural resources. Accordingly, the natural combination turns into an elective strategy for the production of TQ for an enormous scope, with a functional expense contrasted and the segregation of TQ from common sources. In such a manner, a straightforward technique was created for the synthesis of TQ that relies upon catalyzed oxidation of thymol or carvacrol utilizing Co(II) [113]. The most extreme yield of TQ can fluctuate from 84% to 93%, contingent upon the beginning materials (thymol or carvacrol).

A few examinations affirmed that the combination of TQ simple shows a more intense impact in contrast with TQ alone. One of these investigations applied to ovarian malignancy uncovered that analogs of TQ have more potency than TQ [114]. Essentially, the union of TQ-artemisinin hybrids showed a high impact on colon disease [115] and leukemia [116]. Gallate and fluorogallate TQ-analogs showed prevalent efficiency in pancreatic malignancy in vitro [117]. Other studies likewise demonstrated that TQ alone from the volatile oil of N. sativa has a moderate effect against pancreatic malignancy, contrasted with the synthesized TQ-analogs [118]. Collectively, the synthesis of TQ hybrids has a more potent effect on several diseases compared to TQ, and they can overcome the shortage or unavailability of large quantities of pure TQ from natural sources.

## 12. Conclusions

The information in this review provides insight into TQ derived from N. sativa seeds and how it has been demonstrated throughout various investigations to have anticipated beneficial and defensive effects in general, as well as in the context of COVID-19 infection. Due to the dual antiviral action of TQ, combined with its antibacterial, anti-inflammatory, and immunomodulatory properties, TQ is recommended as a highly effective tool in the fight against the novel coronavirus, with significantly lower incidences of side effects. In

addition to its ability to reduce the likelihood of SARS-CoV-2 entry into cells, it also has antiviral properties. Furthermore, the development of TQ nano capsules and analogs has successfully overcome the drawbacks of TQ, as well as its undesirable physical prosperities, allowing for an improvement in the pharmaceutical formulation of TQ for clinical application. It is recommended that additional clinical trials be conducted to investigate the use of TQ in clinical trials.

**Author Contributions:** All authors contributed to collecting the data, drafted or revised the article, gave final approval of the version to be published, and agree to be accountable for all aspects of the work. All authors have read and agreed to the published version of the manuscript.

**Funding:** This research received no external funding.

**Institutional Review Board Statement:** Not applicable.

**Informed Consent Statement:** Not applicable.

**Data Availability Statement:** Not applicable.

**Conflicts of Interest:** The authors declare no conflict of interest.

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
