# Peer review of "COVID-19 and Thymoquinone: Clinical Benefits, Cure, and Challenges"

_2673-8430, doi:10.3390/biomed3010005_

Round 1

Reviewer 1 Report

The manuscript "COVID-19 and Thymoquinone: Current status, Outlook, and Challenges" presented a detailed overview and excellent information on the use of TQ compound in various aspects of Covid-19 and its associated diseases and thus its rising demand. The authors have significantly contributed to describing all potential benefits of TQ in ameliorating different inflammation and disease manifestations of Covid-19 infections. This is a detailed, concise, authentic, and well-written review. The introduction is relevant and informative. Sufficient information about the previous studies and findings and the prevalent association of covid in various manifestations is presented for readers to follow the present study rationale. The inclusion of literature is generally appropriate, although adding a few details of the mechanism of action of TQ in various infections elicited by covid virus and its benefit in many of these immune responses should be provided. Though the review is written well, some modifications in a few sentences and adding details and references could improve it. In my opinion, the manuscript is suitable for publication after the authors have addressed the following comments and questions:

Introduction

1. Authors:- Lines 33-35

Comment: Authors must cite some relevant references here.

2-Authors: Lines- 42-45

Comments: Authors must cite some relevant references for this statement. 

3-Authors: Lines- 45-48: Furthermore, cardiovascular ..with COVID-19 patients. 

Comments: How are these diseases associated with Covid-19; please explain it appropriately, and add a few sentences regarding that. 

2. Are any other studies available, as your reference is insufficient to prove your statement? 

4-Authors: Lines- 50-52: In severe cases.. multiple organ dysfunction and sudden cardiac arrest. 

Comments: Is any statical data available on this, i.e., how many patients/people are affected with covid and show respiratory and cardiac failure? If yes, please mention here which would make this statement more relevant. 

5-Authors: Lines 54-57

Comments: Please rewrite this sentence to follow the readers, as this seems improper to follow the essence of the sentence. 

6-Authors: Lines 58-60: Among the most common ..among other names. 

Comments: Please reframe this sentence. It seems grammatically improper and very hard to follow. 

7-Authors: Lines 61-62

Comments: Could the authors explain what are the dietary purposes? 

8-Authors: Lines 62-64

Comments: 1. Please remove the full stop after reference.

2. What do the authors mean by 30-59-48 percent? It seems inconsistent.

3. What are the biological activities of Nigella sativa? Please mention some brief activities to make this statement more confident. 

4. Please break this sentence into two to make it more straightforward and brief. 

9-Authors: Lines 75-77

Comment: What other treatments could be used to treat covid? Please explain briefly. 

10-Authors: Lines 77-79

Comment: If any other studies reported on the same, please mention them here and cite them. 

11-Authors: Lines 79-81

Comment: Can you please add the immune-checkpoint inhibitors? And from where you gathered this information, please cite some suitable references for these statements.

12-Authors: Lines83-85

Comment: Please describe PDI and anti-PDL1; what are these? This is the first time the authors have mentioned this, and with prior information, it seems easier to follow. 

13-Authors: Lines-87-90

Comments: What do the authors want to describe here? This sentence is challenging to follow. Please rewrite it properly so that readers can follow this information correctly. 

14-Authors: Lines-113-149

Comment: In this section, the authors have described the inflammatory cytokines and chemokines and their role in augmenting the inflammatory response. My question to the authors is, have you read the role of CCL2 chemokine in Covid-infection? As it was recently reported that CCL2 chemokine has a very prominent role in covid-19, could you please add this information to this section? Many studies have reported the CCL2 and its receptor's involvement in covid. Please mention all the relevant information here. 

15-Authors: Lines-140

Comment: Please use a proper style for in-vitro, as the authors have made this mistake throughout the manuscript. 

16-Authors: Lines-156-159

Comment: These sentences need to be clarified for me; please rewrite them accordingly and connect them with the previous information adequately. These sentences need to be more specific to follow. 

17-Authors: Lines 184-186

Comment: Please change the decrease to decreases as it is not grammatically correct. 

18-Authors: lines 229

Comment- What is BV2 microg? Explain it here. Do you mean microglia? Please, rectify these mistakes.

19-Authors: lines 267-268

Comment- What are those tumor markers? Please add some examples here. 

20-Authors: lines 276-277

Comment- What are those ECM proteins? Please add some examples here.

21-Authors: lines 314-316

Comment- TQ shows its antibacterial activity against what kind of bacterial stains; please mention the name of some bacteria to make that sentence appropriate. 

22-Authors: lines 365-367

Comment- What are the harmful effects produced by those compounds on the cells? Mention them briefly. 

22-Authors: Figure 4

Comment- Can you change the color of the TQ circle as TQ drawbacks circles have an almost similar shade to that, and it confuses viewers? 

23-Authors: Conclusion

Comment- Though the authors have explained all sections well, the conclusion should be reasonably debated and consist of more elaborated roles and functions of TQ in covid-19 and other diseases and its rising surge in that context. 

Author Response

Response to the Comments from the editor and reviewers

 First, we thank the editor and reviewers for providing constructive comments and suggestions, which were useful for improving the quality of the manuscript. We have now modified our manuscript in acknowledgement of all of the editor’s and reviewers’ comments and suggestions. The changes made are highlighted in red.

Reviewer 1

The manuscript "COVID-19 and Thymoquinone: Current status, Outlook, and Challenges" presented a detailed overview and excellent information on the use of TQ compound in various aspects of Covid-19 and its associated diseases and thus its rising demand. The authors have significantly contributed to describing all potential benefits of TQ in ameliorating different inflammation and disease manifestations of Covid-19 infections. This is a detailed, concise, authentic, and well-written review. The introduction is relevant and informative. Sufficient information about the previous studies and findings and the prevalent association of covid in various manifestations is presented for readers to follow the present study rationale. The inclusion of literature is generally appropriate, although adding a few details of the mechanism of action of TQ in various infections elicited by covid virus and its benefit in many of these immune responses should be provided. Though the review is written well, some modifications in a few sentences and adding details and references could improve it. In my opinion, the manuscript is suitable for publication after the authors have addressed the following comments and questions:

Response:

Thank you reviewer for your nice words. We have addressed all of your valuable comments and suggestions.

Introduction

  1. Authors:- Lines 33-35

Comment: Authors must cite some relevant references here.

Response:

A relevant reference has been added as suggested  at line 38.

2-Authors: Lines- 42-45

Comments: Authors must cite some relevant references for this statement. 

Response:

A relevant reference has been added as suggested at line 48.

3-Authors: Lines- 45-48: Furthermore, cardiovascular ..with COVID-19 patients. 

Comments: How are these diseases associated with Covid-19; please explain it appropriately, and add a few sentences regarding that. 

Response:

We thank the reviewer for his recommendation. We have explained the relation between cardiovascular disease and COVID 19 in cardiopulmonary protective effect of TQ section at lines (177-181).

  1. Are any other studies available, as your reference is insufficient to prove your statement? 

 Response:

A complete review on the relation between COVID 19 and cardiovascular diseases (Parvu S , Muller K, Dahdal D, Cosmin I, Christodorescu R, Duda-Seiman D, Man D,  Sharma A,  Drago R , Baneu P, Dragan S, COVID-19 and cardiovascular manifestations, European Review for Medical and Pharmacological Sciences 2022; 26: 4509-4519) was published in 2022 and was cited in the manuscript, L565 at references list.

4-Authors: Lines- 50-52: In severe cases.. multiple organ dysfunction and sudden cardiac arrest. 

Comments: Is any statical data available on this, i.e., how many patients/people are affected with covid and show respiratory and cardiac failure? If yes, please mention here which would make this statement more relevant. 

Response:

Statistical data have been added as suggested  at lines (182-189), and subsequently new citation was added as follows;

Shaobo Shi, M D, Mu Qin, M D, Bo Shen, M D. Association of Cardiac Injury With Mortality in Hospitalized Patients With COVID-19 in Wuhan, China, JAMA Cardiol. 2020;5(7):802-810.

Dur-e-N, Rubab T, Sajjad, A. Sudden cardiac death in COVID-19 Patients, Interventional Cardiology 2022 Volume 14.

5-Authors: Lines 54-57

Comments: Please rewrite this sentence to follow the readers, as this seems improper to follow the essence of the sentence. 

Response:

Rewritten as suggested, lines (55-58).

6-Authors: Lines 58-60: Among the most common ..among other names. 

Comments: Please reframe this sentence. It seems grammatically improper and very hard to follow. 

Response:

Rewritten as suggested, at lines (55-59).

7-Authors: Lines 61-62

Comments: Could the authors explain what are the dietary purposes? 

Response:

The dietary purposes of Nigella Sativa have been explained at lines( 59-62).

8-Authors: Lines 62-64

Comments: 1. Please remove the full stop after reference.

Response:

The full stop has been removed.

  1. What do the authors mean by 30-59-48 percent? It seems inconsistent.

Response:

This sentence has been rewritten in a proper essence at lines  (65).

  1. What are the biological activities of Nigella sativa? Please mention some brief activities to make this statement more confident. 

 Response:

Some of the biological activities of Nigella sativa have been mentioned as suggested, at lines (62-64).

  1. Please break this sentence into two to make it more straightforward and brief. 

 Response:

Changed as suggested at lines (62-65).

9-Authors: Lines 75-77

Comment: What other treatments could be used to treat covid? Please explain briefly. 

Response:

This part was rewritten again to explain the proper meaning, at lines (74-76).

10-Authors: Lines 77-79

Comment: If any other studies reported on the same, please mention them here and cite them. Response:  A new citation has been added as suggested at lines(78-84). 

11-Authors: Lines 79-81

Comment: Can you please add the immune-checkpoint inhibitors? And from where you gathered this information, please cite some suitable references for these statements.

Response: 

Some examples of the immune-checkpoint inhibitors have been added and a new citation was included at line 79.

12-Authors: Lines83-85

Comment: Please describe PDI and anti-PDL1; what are these? This is the first time the authors have mentioned this, and with prior information, it seems easier to follow. 

Response:

The full names of PD1 and PDL1 have been added as suggested at lines (81-83).

13-Authors: Lines-87-90

Comments: What do the authors want to describe here? This sentence is challenging to follow. Please rewrite it properly so that readers can follow this information correctly. 

Response:

We have rewritten the mentioned part in proper way at lines (86-87).

14-Authors: Lines-113-149

Comment: In this section, the authors have described the inflammatory cytokines and chemokines and their role in augmenting the inflammatory response. My question to the authors is, have you read the role of CCL2 chemokine in Covid-infection? As it was recently reported that CCL2 chemokine has a very prominent role in covid-19, could you please add this information to this section? Many studies have reported the CCL2 and its receptor's involvement in covid. Please mention all the relevant information here. 

Response:

         The role of CCL2 chemokine in Covid-infection has been added to  the inflammation and multiple organ failure associated with COVID 19  section at lines ( 125-140).

15-Authors: Lines-140

Comment: Please use a proper style for in-vitro, as the authors have made this mistake throughout the manuscript.

Response:

Both in vitro and in vivo wards have been rewritten in a proper style throughout the revised manuscript.

16-Authors: Lines-156-159

Comment: These sentences need to be clarified for me; please rewrite them accordingly and connect them with the previous information adequately. These sentences need to be more specific to follow. 

 Response:

This part has been removed from the manuscript after modification.

17-Authors: Lines 184-186

Comment: Please change the decrease to decreases as it is not grammatically correct. 

Response:

Changed as suggested, line 195.

18-Authors: lines 229

Comment- What is BV2 microg? Explain it here. Do you mean microglia? Please, rectify these mistakes.

Response:

We have corrected “BV2 microg” to be “BV2 microglial”  at line 238.

19-Authors: lines 267-268

Comment- What are those tumor markers? Please add some examples here. 

 Response:

Examples for liver injury and liver tumor markers have been added as suggested at lines (261-264).

20-Authors: lines 276-277

Comment- What are those ECM proteins? Please add some examples here.

 Response:

ECM proteins examples have been added as suggested at line 267 .

21-Authors: lines 314-316

Comment- TQ shows its antibacterial activity against what kind of bacterial stains; please mention the name of some bacteria to make that sentence appropriate. 

Response:

The antibacterial activity of TQ against certain type of bacterial strain has been added at lines (295-296).

22-Authors: lines 365-367

Comment- What are the harmful effects produced by those compounds on the cells? Mention them briefly. 

Response:

The harmful effects produced by TQ metabolites have been explained properly at lines (350-355).

22-Authors: Figure 4

Comment- Can you change the color of the TQ circle as TQ drawbacks circles have an almost similar shade to that, and it confuses viewers?

Response:  

The color of the TQ circle has been changed as suggested, Fig. 4

23-Authors: Conclusion

Comment- Though the authors have explained all sections well, the conclusion should be reasonably debated and consist of more elaborated roles and functions of TQ in covid-19 and other diseases and its rising surge in that context. 

Response:

We acknowledge the reviewers suggestion. However, in the initial submitted MS, we already mentioned the role of TQ in COVID-19 “L424-427 revised MS”. Owing to the main aim of this review is discussion the role of TQ on cure the complications associated with COVID-19 and how overcoming the challenges faced the clinical application of TQ, we focused on this point on the conclusion at Lines 427-429.

Reviewer 2 Report

The authors of the article entitled " COVID-19 and Thymoquinone: Current status, Outlook, and 2 Challenges" attempt to summarize and discuss publications on thymoquinone beneficial effects in management of COVID-19. Although it is an interesting topic, the authors failed to add significant insights to the field, and most of the information provided is published in a similar way elsewhere. I believe this article could be of general interest if the authors consider the following comments.

1.The author should make the title a bit clearer to accurately reflect the content of the article, especially after the inclusion of the recommended changes.

2- At least, two similar review articles were recently published by Badary et al 2021 in “Drug Des Deve Ther” PMID: 33976534 and Khazdair et al 2021 in Pharm Biol  PMID: 34110959.  These articles should be cited and authors are  strongly advised to update their article and make it significantly distinguishable.  

3- Some of the information presented in the article is not COVID-19 -specific. However, the authors extrapolated the data to the COVID-19 and made several assumptions. It is strongly recommended to indicate the type of experiments (animals or Human) and provide more bases for extrapolation to COVID-19.

4. Authors may summarize the article to avoid the general information of thymoquinone studies that have been widely published and concentrate on the insights on relation to COVID-19 management and treatment

5. It may be very helpful to include and compare the effects of other natural products of promising effects against COVID-19

6-The authors should specify the type of the literature review, decide on a specific design to write the article, and present objective conclusions based upon the literature reviewed. The authors should also mention the sources, search strategy, and selection criteria used to collect the information presented in this article.
7. In the abstract, please mention your take home message from this article. A  recommendation statement at the end of the abstract is also needed. The current abstract may be altered to accommodate more relevant information.

Author Response

Response to the Comments from the editor and reviewers

 First, we thank the editor and reviewers for providing constructive comments and suggestions, which were useful for improving the quality of the manuscript. We have now modified our manuscript in acknowledgement of all of the editor’s and reviewers’ comments and suggestions. The changes made are highlighted in red.

Reviewer 2:

The authors of the article entitled " COVID-19 and Thymoquinone: Current status, Outlook, and Challenges" attempt to summarize and discuss publications on thymoquinone beneficial effects in management of COVID-19. Although it is an interesting topic, the authors failed to add significant insights to the field, and most of the information provided is published in a similar way elsewhere. I believe this article could be of general interest if the authors consider the following comments.

Response:

We thank the reviewer for their valuable comments and suggestion. We have addressed all the suggested comments.

1.The author should make the title a bit clearer to accurately reflect the content of the article, especially after the inclusion of the recommended changes.

Response:

We have changed the title to be more specific for the content of the manuscript, Line 2.

2- At least, two similar review articles were recently published by Badary et al 2021 in “Drug Des Deve Ther” PMID: 33976534 and Khazdair et al 2021 in Pharm Biol  PMID: 34110959.  These articles should be cited and authors are  strongly advised to update their article and make it significantly distinguishable.  

Response:

We thank the reviewer for his recommendation. The two articles have been cited in the revised manuscript at lines (217-224) and (337-341).

3- Some of the information presented in the article is not COVID-19 -specific. However, the authors extrapolated the data to the COVID-19 and made several assumptions. It is strongly recommended to indicate the type of experiments (animals or Human) and provide more bases for extrapolation to COVID-19.

Response:

We have indicated the type of experiments. In addition, we have provided more bases for extrapolation to COVID-19, lines (125-139), (148-152), (166-171), (182-189), (202-207) and (224-227).

  1. Authors may summarize the article to avoid the general information of thymoquinone studies that have been widely published and concentrate on the insights on relation to COVID-19 management and treatment

Response:

We have summarized the article in different sections and avoid general information of thymoquinone studies.

  1. It may be very helpful to include and compare the effects of other natural products of promising effects against COVID-19

Response:

The effect of some natural compound on the COVID 19 entry and replication inside the host cell have been added at lines (303-311).

6-The authors should specify the type of the literature review, decide on a specific design to write the article, and present objective conclusions based upon the literature reviewed. The authors should also mention the sources, search strategy, and selection criteria used to collect the information presented in this article.

Response:

 A method section has been added to illustrate the sources, search strategy, and selection criteria used to collect the information presented in this article at lines (108-114)

  1. In the abstract, please mention your take home message from this article. A  recommendation statement at the end of the abstract is also needed. The current abstract may be altered to accommodate more relevant information.

Response:

We acknowledge the reviewer’s comment. The home message from this article was mentioned in the initial submitted MS “L25-28, revised MS”. For the recommendation statement we have added at the end of the abstract, L28-32.

Reviewer 3 Report

1-      Please define all abbreviations at the first uses in the text.

2-      Please provide methods section.

3-      Please provide more literature about protective effects (anti-inflammatory effect) of TQ on lung inflammation.

4-      Please provide more literature about antioxidant effect of TQ on clinical studies.

5-      Please define “TnT”.

6-      Please summarized the results of the manuscript in appropriate Tables.

7-      Please reduce the numbers of references.

8-      It is competent to use the below relevant studies in different sections.

A qualitative and quantitative comparison of Crocus sativus and Nigella sativa immunomodulatory effects. Biomedicine & Pharmacotherapy140, p.111774.

The possible therapeutic effects of some medicinal plants for chronic cough in children. Evidence-based complementary and alternative medicine2020.

Thymoquinone ameliorates lung inflammation and pathological changes observed in lipopolysaccharide-induced lung injury. Evidence-based Complementary and Alternative Medicine. 2021 Mar 30;2021.

Possible therapeutic effects of Nigella sativa and its thymoquinone on COVID-19. Pharmaceutical biology. 2021 Jan 1;59(1):694-701.

Author Response

Response to the Comments from the editor and reviewers

 First, we thank the editor and reviewers for providing constructive comments and suggestions, which were useful for improving the quality of the manuscript. We have now modified our manuscript in acknowledgement of all of the editor’s and reviewers’ comments and suggestions. The changes made are highlighted in red.

Reviewer 3:

1-Please define all abbreviations at the first uses in the text.

Response:

All abbreviations have been defined as suggested.

2-Please provide methods section.

Response:

Method section has been included as suggested, Lines (108-114).

3-Please provide more literature about protective effects (anti-inflammatory effect) of TQ on lung inflammation.

           Response:

More literatures have been added as suggested at lines (144-148) , (214-224) and (236-238).

4- Please provide more literature about antioxidant effect of TQ on clinical studies.

 Response:

The antioxidant effect of TQ on clinical studies was discussed in the Oxidative stress associated with COVID 19 and the antioxidant effect of TQ section of the manuscript at lines (163-174).

5- Please define “TnT”.

 Response:

TnT was defined as suggested, line 195.

6- Please summarized the results of the manuscript in appropriate Tables.

Response:

The results of the manuscript were summarized in an appropriate table as suggested, Table  1.

7- Please reduce the numbers of references.

Response:

We have reduced the number of references from 155 references in the initial submitted MS to 118 references in the revised MS.

8- It is competent to use the below relevant studies in different sections

  Response:

Two relevant studies were added as suggested, at lines (216-217) and (222-224) and subsequently two new citation were included as following;

Boskabady, M.; Khazdair, M.R.; Bargi, R.; Saadat, S.; Memarzia, A.; Mohammadian Roshan, N.; Hosseini, M.; Askari, V.R.; Boskabady, M.H. Thymoquinone Ameliorates Lung Inflammation and Pathological Changes Observed in Lipopolysaccharide-Induced Lung Injury. Evid Based Complement Alternat Med 2021, 2021, 6681729, doi:10.1155/2021/6681729.

Khazdair, M.R.; Ghafari, S.; Sadeghi, M. Possible therapeutic effects of Nigella sativa and its thymoquinone on COVID-19. Pharm Biol 2021, 59, 696-703, doi:10.1080/13880209.2021.1931353.

Round 2

Reviewer 1 Report

The authors have mentioned and incorporated all the relevant information and resolved my concerns. 

Reviewer 3 Report

31 Dec 2022

Dr. Vickie Xie Assistant Editor BioMed

E-Mail: [email protected]

 Dear Dr. Vickie Xie

Thank you for your Email regarding to review the revised version of the Manuscript titled " COVID-19 and Thymoquinone: Clinical benefits, Cure, and Challenges " The manuscript is suitable for publication in the present form.

Yours sincerely